# A Novel Monopolar Cross-Scale Nanopositioning Stage Based on Dual Piezoelectric Stick-Slip Driving Principle

**DOI:** 10.3390/mi13112008

**Published:** 2022-11-18

**Authors:** Junhui Zhu, Siyuan Meng, Yong Wang, Ming Pang, Zhiping Hu, Changhai Ru

**Affiliations:** 1School of Mechatronic Engineering and Automation, Shanghai University, Shanghai 200444, China; 2State Key Laboratory of Robotics and System, Harbin Institute of Technology, Harbin 150000, China; 3School of Electronic and Information Engineering, Suzhou University of Science and Technology, Suzhou 215000, China; 4College of Intelligent Systems Science and Engineering, Harbin Engineering University, Harbin 150001, China; 5Micro-Nano Automation Institute, JITRI, Suzhou 215100, China

**Keywords:** nanopositioning, stick-slip, dual piezoelectric drive, flexible hinge, compact structure

## Abstract

The precise characterization and measurement of new nanomaterials and nano devices require in situ SEM nanorobotic instrumentation systems, which put forward further technical requirements on nanopositioning techniques of compact structure, cross-scale, nanometer accuracy, high vacuum and non-magnetic environment compatibility, etc. In this work, a novel cross-scale nanopositioning stage was proposed, which combined the advantages of piezoelectric stick-slip positioner and piezoelectric scanner techniques and adopted the idea of macro/micro positioning. A new structure design of a single flexible hinge shared by a small and large PZT was proposed to effectively reduce the size of the positioning stage and achieve millimeter stroke and nanometer motion positioning accuracy. Then, the cross-scale motion generation mechanism of the dual piezoelectric stick-slip drive was studied, the system-level dynamics model of the proposed positioning stages was constructed, and the mechanism design was optimized. Further, a prototype was manufactured and a series of experiments were carried out to test the performance of the stage. The results show that the proposed positioning stage has a maximum motion range of 20 mm and minimum step length of 70 nm under the small piezoceramic ceramic macro-motion stepping mode, and a maximum scanning range of 4.9 μm and motion resolution of 16 nm under the large piezoceramic ceramic micro-motion scanning mode. Moreover, the proposed stage has a compact structure size of 30 × 17 × 8 mm^3^, with a maximum motion speed of 10 mm/s and maximum load of 2 kg. The experimental results confirm the feasibility of the proposed stage, and nanometer positioning resolution, high accuracy, high speed, and a large travel range were achieved, which demonstrates that the proposed stage has significant performance and potential for many in situ SEM nanorobotic instrument systems.

## 1. Introduction

With the continuous discovery and recognition of new nanomaterials and nano devices with excellent chemical, electrical and mechanical properties, these materials have become of significant interest and impact for the development and innovation of biomedicine, new energy batteries, new generation semiconductors, and many other fields [1,2,3,4]. The precise characterization and measurement of these new nano materials and devices are challenging tasks that need in situ SEM (scanning electron microscope) nanorobotic instrument systems [5,6,7,8]. These systems, such as the nanomanipulation system, AFM (atomic force microscope), and nanoindentation instrument, etc. [9], are required to move, manipulate, and characterize micro-nano objects in the SEM of compact and vacuum environments. One of the key technical challenges of the nanorobotic systems is the need for cross-scale nanopositioning technology with a compact structure and compatible with a high vacuum environment. Therefore, cross-scale nanopositioning with millimeter stroke, nanometer resolution and positioning accuracy, and large motion velocity has been a hot research topic. Currently, technologies that can achieve cross-scale nanopositioning with millimeter stroke and nanometer resolution include the macro-micro dual drive principle, inchworm drive principle, a piezoelectric ultrasonic motor, and stick-slip drive principle [10,11,12,13,14,15,16,17]. Among them, the size of the inchworm drive principle structure is larger, and control is more complicated, and it is not easy to integrate into the SEM [18]. The stick-slip drive principle has the advantages of high resolution, compact structure, large displacement stroke, vacuum compatibility, no electromagnetic interference, and no heat generation [19]. However, due to its own technical characteristics, the difference between the sliding friction force and maximum static friction force of the contact surface forms multiple cycles to control the displacement of the object, which is mostly used for positioning between two points. During this process, jitter is generated. After the actuator is amplified, it even generates a micrometer-level amplitude, and cannot perform nanometer-level steady-state motion. Similarly, the piezoelectric ultrasonic motor utilizes vibration to generate displacement, which is also easy to cause this phenomenon [20]. The macro-micro dual drive principle is composed of two motion states: fast “macro movement” and precise “micro movement”, which makes this technical form have a long movement range and fast response speed, and has attracted the attention of many scholars. A macro-micro dual-drive high-acceleration, high-precision XY positioning stage was proposed by Jie D et al. [21]. Macro motion uses a VCM (voice coil motor) to provide power input, and micro motion uses a PZT (piezoelectric ceramic transducer) to provide power input, which realizes high-speed movement in a space of 25 × 25 mm^2^, where acceleration exceeds 100 mm/s^2^ and the positioning accuracy in X and Y directions is better than 10 nm. However, the volume of this program is too large, according to the article that speculates on the choice of VCM; its size should be more than 150 × 150 × 100 mm^3^, and its “macro movement” state is provided by VCM. It is easy to generate heat and generate electromagnetic interference, and it is not compatible with SEM. Many scholars have turned the macro-movement to the stick-slip drive principle, using the idea of combining macro-micro dual drive motion and the stick-slip drive principle instead of VCM drive, providing more possibilities for cross-scale nano-motion positioning. A two-dimensional manipulation stage driven by two PZT actuators in parallel was proposed by Yuki Shimizu et al. [22]. The straightness of the stage movement was ensured by a flexible piece, and a permanent magnet was used as the slider, which was a good solution to the problem of coupling interference of two-dimensional movement in the same driving plane. The stage can realize two-dimensional motion along the X-Y direction, and motion speed can reach 5 mm/s, with the size of 24 × 24 × 5 mm^3^, but effective strokes in the X and Y directions are only ±1 mm. An X and Y two-dimensional nanopositioning stage with large motion stroke and high resolution was proposed by Fujun Wang et al. [23]. They used a low coupling rate based on stick-slip drive using parallel decoupled drive units and a tandem configuration of sliders to achieve hybrid output decoupling. The bottom surface of the stage was a flexible hinge with X and Y two-dimensional coupling drive, and the PZTs were arranged as drive elements at the center of the four sides, respectively, and a jacking device was set up at the center of the stage to realize the decoupling of the motion in both directions and make the overall mechanism more compact. A simple, adjustment-free, and open-source XYZ nanopositioner developed by Hsien-Shun Liao et al. [24] utilized a magnet-based driving mechanism. The magnet simultaneously provided both a preload force and friction surface, resulting in a nanopositioner with compact size, low cost, easy assembly, and high-vacuum compatibility. The macro-micro dual drive principle requires the combination of two motion states; that is, two or more motion actuators are required, resulting in a large structure with a single-degree-of-freedom motion axis. Therefore, many scholars targeted their research to the two-level macro and micro motion combined nanopositioning stage, which can realize the macro motion of large strokes and micro motion of nanometer positioning resolution by delicate structures that separate and then couple the macro and micro motions. Though SEM compatible nanorobotic instrument systems are usually composed of one or more three-degree-of-freedom XYZ axis nanomanipulators, the above-mentioned traditional two-level macro and micro motion combined nanopositioning stage will lead to problems such as complex structure and large volume, control complexity, and poor adaptability of SEM of different vacuum chamber sizes and models. Therefore, monopole kinematic positioning technique compatible with a compact, cross-scale, nanopositioning, high vacuum, and non-magnetic environment is needed to solve the drawback stage.

In this work, a novel cross-scale nanopositioning stage, based on a dual piezoelectric stick-slip driving stage principle is proposed, which applies the idea of the combined macro and micro motion positioning and stick-slip driving principle, by utilizing a small and large PZT as the “macro” and “micro” driver, respectively. The advantages of various cross-scale nano-motion positioning techniques, such as macro-micro, stick-slip, and piezoelectric scanning stages, are combined to achieve compact structure, cross-scale large stroke, and nanopositioning on a single axis. By studying the cross-scale motion generation mechanism of dual piezoelectric stick-slip drive, a system-level dynamics model of the motion positioning stage was constructed and the design of its mechanism was optimized. Finally, experiments were conducted on a prototype of the developed monopolar cross-scale nanopositioning stage. Nanometer positioning resolution, high accuracy, high speed, and large travel range were achieved, demonstrating that this stage has significant performance and potential in many in situ SEM nanorobotic instrument systems.

## 2. Principles of Movement

The movement principle of the dual piezoelectric stick-slip driving positioning stage based on sawtooth wave drive signals is shown in Figure 1a. The small PZT is responsible for driving the “macro” fast motion using the stick-slip drive method, and the large PZT is responsible for the “micro” precise positioning using its own scanning mode.

“Macro-motion” mode: as shown in Figure 1b, the t0 to t1 phase applies a slowly rising voltage to the small PZT, which slowly elongates to form a displacement, and the drive mechanism uses friction to drive the rail to move. The t1 to t2 phase gives a step-down voltage signal to the small PZT, which makes it quickly retract. The inertia force is much larger than the friction force, so the rail remains motionless; the small PZT returns to its original position, and the rail forms a displacement. Multiple t0 to t2 cycles of motion forms macro-displacement $X_{h}=\sum_{1}^{n}\left(x_{1}\right)$ to achieve rapid motion across the scale.

“Micro-motion” mode: as shown in Figure 1b, a constant voltage is applied to the large PZT from t3 to t4 to make it enter the scanning mode, which will slowly elongate and achieve nanoscale motion and positioning.

## 3. Design and Analysis

### 3.1. Design of Overall Structure

Figure 2 schematically shows the structure of the macro-micro-combined dual piezoelectric stick-slip nanopositioning stage. It mainly consists of a base, drive unit, guide, and various screws. The drive unit consists of a large and small PZT with different respective nominal displacements, micro and macro, and a frictionless, backlash-free, and highly responsive flexible hinge connected in series with each other [25,26,27]. Among them, the upper surface of the flexible hinge with a friction plate in contact with the rail uses the stick-slip drive mechanism to generate displacement by adjusting the friction force screw to change the positive pressure between the friction plate on the flexible hinge and the rail. This ensures full contact between the friction plate and rail with a uniform force. The drive unit is installed in the prefabricated groove of the base, and the preload screw is adjusted to change the preload force between the drive unit and base; the guide rail and base are connected by multiple mounting screws in the vertical direction.

When a small PZT sawtooth wave drive voltage is given, it is transmitted to the guide by a flexible hinge using static friction, and then quickly resumes its reciprocating periodic motion, whereas when a large PZT constant drive voltage is given, it is transmitted directly by a flexible hinge using static friction. Therefore, the generation and transmission of “macro-motion” and “micro-motion” are realized by a single system, which ensures cross-scale nano-motion positioning while significantly reducing the size of the nano-positioning stage. This would lead to the possibility of integrating the nanomanipulator and AFM with the SEM.

### 3.2. Drive System Modeling and Analysis

The output displacement of the dual PZT actuated positioning stage is achieved by the interaction between the PZT actuator and the flexible hinge. The PZT acts on the flexible hinge, and the flexible hinge reacts to the PZT, and the output displacement of the PZT is different from its free displacement. Therefore, it is necessary to study the dynamics model of the drive train system, which is a combination of the PZT actuator and flexible transmission parts. The dynamics model of the positioning stage consists of three parts: the electrical model of the PZT, dynamics model of the drive train composed of the PZT and flexible hinge, and the dynamic friction model, as shown in Figure 3. The model is applied to analyze the influence of the design parameters of each part on the positioning stage. The main parameters include the equivalent mass of the flexible hinge *m_i_*, equivalent damping of the flexible hinge *c_s_*, equivalent stiffness of the flexible hinge *k_s_*, positive pressure *F_N_* and mass of the guide *m_s_*, equivalent mass of PZT *m_p_*, equivalent damping of PZT *c_p_*, equivalent stiffness of PZT *k_p_*, output displacement of the flexible hinge *x*, and output displacement of the nanopositioning stage *x_s_*.

The PZT is driven by the voltage signal, then the electromechanical conversion model converts the voltage into output force and output displacement, and the signal is transmitted to the PZT and flexible hinge, which is then transferred to a drive–driven second-order oscillation system consisting of a piezoelectric ceramic and flexible hinge. Finally, the signal goes through the conversion of the dynamic and static friction between the flexible drive components and then drives the guide to achieve periodic motion. The system dynamics model transfer flow chart as shown in Figure 4a. Based on the transfer function of each model, we built a simulation model using MATLAB, as shown in Figure 4b. The above design parameters were further verified and compared using the control variable method, and the simulation results are shown in Figure 5.

Based on the analysis of the above simulation results, we found that the larger the equivalent mass of the flexible hinge, the smaller the output displacement of the positioning stage, and the more obvious the oscillation phenomenon. The larger the equivalent stiffness of the flexible hinge, the smaller the output displacement of the positioning stage; the smaller the equivalent stiffness, the worse the dynamic response characteristics of the drive transmission system. Within a certain range, increasing the friction force by adjusting the positive pressure could increase the effective output displacement of the positioning stage and improve it. However, beyond a certain range, the guide rail was a serious obstacle to the drive transmission system, reducing the effective output displacement of the stage, causing even zero motion speed; the guide rail was equivalent to the load applied to the positioning stage, and the larger the mass of the guide rail, the larger the positive pressure, resulting in the inability of the flexible transmission parts to completely deform, and a smaller displacement transferred to the guide rail. Overall, these findings provide strong support for the integration of the positioning stage, and we selected the optimal data for the design and integration of the positioning stage.

### 3.3. Design of Flexible Hinge Structure

The drive unit consists of a flexible hinge shared by the two PZTs driving the “macro” and “micro” movements. Thus, the flexible hinge needs to have a certain degree of versatility in a small area, and then analyze the force relationship between the selected PZT and the flexible hinge based on the following parameters of the selected PZT.

Figure 6 shows the relationship between the hysteresis force and flexibility of the flexible hinge curve; the manufacturer recommends the optimal preload force of 50 N. When the displacement of the large PZT is 5.5 μm, the displacement of the small PZT 1.375 μm, in line with the “macro” and “micro” movements. The analysis shows that the best flexibility of the two PZTs corresponding to the flexible hinge is 0.11 and 0.02 μm/N, respectively, so the flexibility of the flexible hinge is 0.11 μm/N (i.e., the minimum stiffness is 9.09 μm/N).

According to the requirements of the above analysis, the flexible hinge mechanism needs to have a preload hinge with adjustable preload force direction, a friction hinge with adjustable friction force direction, and an overall drive hinge, which also needs to match the size of the two ceramics and have friction plates; its specific structure is shown in Figure 7.

Due to the application of the preload force, the flexible hinge experiences different phases of motion, which is simplified into a hinged four-bar mechanism, as shown in Figure 8a. Figure 8b shows the deformation state when the output force is less than or equal to 50 N of preload force, at which time $x_{1}$ is the preload deformation variable of the flexible transmission member; Figure 8c shows the deformation state when the output force is greater than a 50 N preload force, and$\;x_{2}-x_{1}\;$is the deformation variable of the flexible transmission member when the output force is applied. Until the output force of the piezoelectric stack reaches the maximum value of 160 N, the deformation variable occurring in the flexible transmission member also reaches the maximum value, i.e., $x_{2}-x_{1}$ reaches the maximum value.

### 3.4. Analysis and Simulation of Flexible Hinge Structure

The following three main factors affect the performance of the flexible hinge: the stiffness, strength, and resonant frequency of the flexible hinge.

Stiffness problem. Too little stiffness will increase the deformation under the preload force affecting its life, reducing return speed, and lowering the dynamic response speed; too much stiffness will increase the resistance to the piezoelectric stack, and the output displacement transferred to the guide will be reduced, lowering the movement speed of the positioning stage.Strength problem. Flexible hinges mainly use the small deformation and self-return characteristics generated by elastic materials to improve the displacement resolution; too little strength will cause plastic deformation or even fatigue damage due to alternating loads.Resonant frequency problem. The PZT achieves periodic vibration under the excitation of a sawtooth wave voltage signal. If the working frequency of the PZT is greater than or equal to the frequency of any mode of the flexible hinge, it will cause the flexible hinge to produce a self-excitation or resonance phenomenon with the PZT, resulting in unstable stage motion.

Therefore, the above three factors need to be simulated and analyzed to verify whether the flexible hinge meets the requirements.

#### 3.4.1. Stiffness Analysis

As the flexible hinge and the two PZTs have the same direction of transmission and drive, they can be regarded as an integrated structure. Therefore, the minimum stiffness of the two piezoelectric ceramics is 20 N/μm according to Table 1, which can be the stiffness standard of the flexible hinge. In this work, phosphor bronze was selected as the flexible hinge material because of its small density, good wear resistance, and high strength, among other advantages. The results of the stiffness simulation analysis are shown in Figure 9. When a 50 N horizontal thrust was applied to the flexible hinge at the contact surface, the output displacement of the end of the flexible transmission member in the horizontal direction was 5.5767 μm. Calculated according to the simulation results, the stiffness of the transmission flexible hinge was 8.966 N/μm, which was less than the stiffness standard of the flexure hinge with 20 N/μm, and met the design requirements and ensured that the output displacement of the piezoelectric stack could be quickly transferred to the guide rail through the flexible transmission member with minimum displacement loss.

#### 3.4.2. Strength Analysis

The horizontal deformation of the flexible hinge is caused by both the preload and output force of the PZT. The maximum output displacement of the PZT is 8 μm, and preload deformation is 5.7 μm, so the maximum output displacement is 13.7 μm. A 13.7 μm horizontal displacement constraint is applied at the horizontal force surface of the flexible hinge, and the simulation results are shown in Figure 10. The maximum stress value is 156 MPa, the safety factor is 1.2, safety yield strength is about 160 MPa, and yield strength of phosphor bronze material is 193 MPa. Thus, the flexible hinge met strength and safety requirements. It was verified that the flexible hinge could be safely used at rest and at work without plastic deformation and fatigue fractures due to the preload force or PZT output force.

#### 3.4.3. Modal Analysis

Using the finite element simulation software ANSYS/Workbench to analyze the modal vibration pattern of the flexible transmission parts, the first six order vibration patterns of the hinge were obtained, as shown in Figure 11, and the frequencies from the first order mode to the sixth order mode were: 31,667, 53,205, 75,848, 83,849, 88,991, and 91,094 Hz, respectively. The simulation results show that the operating frequency of the PZT drive power supply should be controlled in a frequency range much less than 31,667 Hz.

## 4. Experimental Results

The nanopositioning stage was manufactured by EDM with a machining resolution better than 0.5 mm. The material of the base and stage was 6061Al, the cross roller guide rail was model NB SVWS1030 with a stroke of 20 mm, and the material of the flexible hinge was phosphor bronze. The overall size was 30 × 17 × 8 mm^3^, as shown in Figure 12. The large PZT size was 2 × 2 × 8 mm^3^, with a nominal displacement of 8 μm, and the small PZT size was 2 × 2 × 2 mm^3^, with a nominal displacement of 2 μm, both with a maximum thrust force of 100 N. The sawtooth wave was generated by a DAC chip TLV5619, and after amplification, the signal could continuously output a current of 350 mA at an amplitude of 100 V. The displacement of the positioning stage was measured by a non-contact laser interferometer (POLYTEC OFV3001), placed on the vibration isolation table. The data was read and the displacement curve is displayed by the LK-Navigator software.

First, the “micro-motion” function of the nanopositioning stage was tested, including the range of motion and resolution.

Range of motion: The voltage applied by the large PZT was taken from 40 to 90 V. The test was repeated eight times, as shown in Figure 13a. The average value of the range of motion of the micromotion was 4.9 μm, which was different from the theoretical value of 5.5 μm, and the analysis was due to two reasons: (1) The PZT actuator acted on the flexible transmission and was also subject to its own reaction force, which hindered the output of the PZT displacement. (2) The output displacement of the PZT is transmitted to the guide through the drive transmission system and through the action of friction, which causes a loss of output displacement due to the presence of friction.

Resolution: The experiment applied a sine wave signal to the positioning stage to test the resolution of the scanning mode of the positioning stage, setting the frequency at 1 Hz. The voltage started from 0 V and gradually increased with a gradient of 0.1 V until the voltage amplitude reached 0.6 V and the stage displacement output was stable, as shown in Figure 13b. The measured data were fitted to obtain the motion resolution of the positioning stage in the scanning mode of about 16 nm, which was better than the traditional piezoelectric stick-slip positioning stages, such as the stick-slip stage developed by Rakotondrabe et al., with resolution of 70 nm [28], and the commercialized stick-slip stage of model ANP_X_101 developed by Attocube System Inc., with a resolution of about 200 nm [29].

Next, the “macro motion” function of the nanopositioning stage was tested, including the maximum speed, maximum step length, minimum step length, maximum thrust, and speed consistency in the horizontal motion direction. Before the step and speed test, we needed to test the influence of different driving signals (driving frequency, voltage amplitude, etc.) on the stick-slip drive, and the experimental results are shown in Figure 14. Eight groups were realized, and finally, the average value of the velocity was taken and plotted as the relationship curve in Figure 14c. The output displacement curve of the positioning stage under the action of the voltage signal with the driving frequency of 50 and 100 Hz is shown in Figure 14a,b, respectively. When studying the relationship between voltage amplitude and positioning stage movement speed, the voltage amplitude was taken as 40, 50, 60, 70, 80, and 90 V, and the frequency of the driving signal was set as 1 KHz. Eight sets of experiments were conducted, and the average value of speed was finally drawn as Figure 14d. Analysis shows that the greater the frequency of the driving signal, the faster the motion frequency of the nanopositioning stage, and the greater the voltage amplitude, the longer the step length of the nanopositioning stage. The nanopositioning stage macro motion speed = step length * frequency, so any increase in either will cause the speed of the nanopositioning stage to increase. When the driving voltage signal amplitude was 90 V and frequency was 15 KHz, the maximum motion speed of the positioning stage in the horizontal direction was found to be 10 mm/s through several tests.

Maximum step length: The maximum drive voltage of 90 V and the 1 Hz sawtooth wave voltage of the small PZT were selected for experimental testing, and the step displacement curve was measured by the KEYENCE laser interferometer, as shown in Figure 15a. After 10 cycles of signal driving, the output displacement in the forward direction was about 9 μm; that is, the step displacement in each cycle was about 0.9 μm. We further inputted a sawtooth wave signal with a frequency of 1 Hz and amplitude of 90 V to the simulation model of Figure 4 to verify the correctness of the simulation method, as shown in Figure 15b. To analyze the effective output displacement of the positioning platform more intuitively, we further derived the simulation curve of the effective output displacement of the positioning stage, and the maximum average step size of the simulation result was 1.375 μm, as shown in Figure 15c. The correctness of the simulation model was verified.

The minimum step: We applied a sawtooth wave voltage signal with a frequency of 1 Hz to the small PZT, and the initial voltage amplitude was set to 90 V. We reduced the size of the input voltage signal, in turn, until the minimum output displacement could be detected. When the minimum driving voltage of 10 V was detected, this output displacement was the minimum movement of the positioning stage. The displacement of 10 steps forward and 10 steps in reverse were 700 and 710 nm, respectively, as shown in Figure 15e,f. After calculation, the minimum step length forward and reverse was 70 and 71 nm, respectively. The displacement was more symmetrical and linearity was better.

Maximum thrust: This is one of the important indexes to measure the performance of the positioning stage. Here, the electronic scale was chosen to measure the thrust, the maximum range was 2 kg, and the index value was 0.01 g. As shown in Figure 15d, at the beginning of the test, we ensured that one end of the positioning stage touched the electronic scale pallet. The positioning stage moved down to make electronic readings, and when the number displayed by the electronic scale reached the maximum and remained unchanged, the force at this time was taken as the maximum thrust. After ten tests, the average value of the maximum thrust was 1.2 N.

Horizontal movement direction speed consistency: To detect the consistency of the forward and reverse movement speed of the positioning stage, as well as the difference, the positioning stage was driven by 50 Hz, with a 90 V sawtooth wave signal, and eight groups of forward movement data and eight groups of reverse movement data were obtained, as shown in Figure 15g. The experimental results are shown in Figure 15h; the speed consistency of the positioning stage in both forward and reverse motion was good. The reverse motion speed was slightly higher than the forward motion speed, mainly because the friction force between the guide rail and the flexible transmission parts was different in the forward and reverse motion of the positioning stage.

At the end, we also performed a load test of the nanopositioning stage. By adding different masses of weights on the guide rail, the masses were classified as 400 g, 800 g, 1.2 kg, 1.6 kg, and 2 kg. Setting the voltage amplitude of the driving signal to 90 V and the frequency to 1 KHz, the output displacement of the positioning stage in step mode at different loads was measured, and eight sets of data were obtained for each load quality experiment, as shown in Table 2. The output displacement of the positioning stage is plotted according to the data in Figure 16 and shows the relationship between the output displacement of the positioning stage and load quality.

In order to more clearly express the influence of the load mass on the motion performance of the positioning stage in the horizontal direction, the average value of the output displacement of the positioning stage under each load mass was plotted as a histogram in Figure 16b. With an increase in the load mass, the output displacement of the positioning stage gradually decreased, and when the load mass increased from 1200 to 2000 g, the output displacement of the stage decreases from 0.7 to 0.28 μm, and the reduction was obvious. As there was a gap between the roller cage in the guide and the V-slide when the positioning stage was not subjected to a load, the output displacement decreased as the friction between the guide and flexible hinge increased with an increase in load. The forward motion speed of the positioning stage was collated with the load mass, as shown in Figure 16c. With an increasing load, the speed of the positioning stage gradually decreased, and when the load mass was 2 kg, the speed of the positioning stage was only 0.28 mm/s, so the maximum load capacity of the positioning stage was 2 kg, and the error was controllable. Continuing to increase the load would lead to a decrease in speed and difficult movement of the positioning stage.

## 5. Discussion and Conclusions

In order to meet the needs of advanced nanomaterials and structures in situ, researchers have used nano-manipulation and inspection instruments for precise and stable nanometer-scale motion accuracy, millimeter-scale motion stroke, vacuum non-magnetic compatibility, and compact form factor nano-positioning technology. We proposed a dual piezoelectric nano-motion positioning method using macro-micro combination, and a new compact cross-scale macro-micro combination dual piezoelectric stick slip monopole nano-motion positioning stage was constructed. The size of the nano-positioning stage was significantly reduced by integrating dual PZTs sharing a single flexible hinge mechanism. The experimental results showed that the positioning stage had a resolution of 16 nm in micro-motion mode, maximum micro-motion range of 4.9 μm, maximum step in macro-motion mode of 0.9 μm, minimum step of 70 nm, maximum motion speed of 10 mm/s, maximum load of the positioning stage of 2 kg, and maximum motion range of 20 mm for the positioning needs of in-situ nano-operating instruments, such as nano-operating machines and AFMs compatible within the SEM. They provide a compact, cross-scale, nanoscale motion positioning, and vacuum non-magnetic compatible single-degree-of-freedom motion positioning technology, giving a new monopole cross-scale nano-motion and positioning compact solution.

It is noteworthy that the small PZT could output a displacement of 1.375 μm without preload, but the actual detection was only 0.9 μm. This is to ensure that the small PZT was not damaged during operation, so a certain size of preload was applied to it in the initial state to eliminate the gap between the ceramic and flexible transmission parts, so that the small PZT and flexible transmission parts were combined to form a second-order oscillation system. The output displacement was reduced under the action of the preload force. Secondly, the actual output displacement of the PZT was smaller than theoretical values due to the existence of the equivalent stiffness of the flexible hinge, as mentioned above.

There are still some difficulties in the control and integration of the current nanopositioning stage, which should be addressed in future work. First, the “macro” and “micro” modes of the positioning stage are time-sharing and segmented control, and future work needs to softly control them, so that when a fixed position is given, the macro and micro movements can be smoothly and automatically switched through autonomous calculations, and the target point can be accurately and stably reached without jitter. Second, the positioning stage is currently an open-loop system with local step inconsistency. In the next study, we will consider designing the positioning stage as a closed-loop positioning system, choosing the scale as the displacement sensor to achieve real-time feedback of the displacement of the positioning stage. Finally, the positioning stage should be open to be compatible with multiple forms of mechanical interfaces to facilitate integration with a variety of nanopositioning stages.

## Figures and Tables

**Figure 1 micromachines-13-02008-f001:**
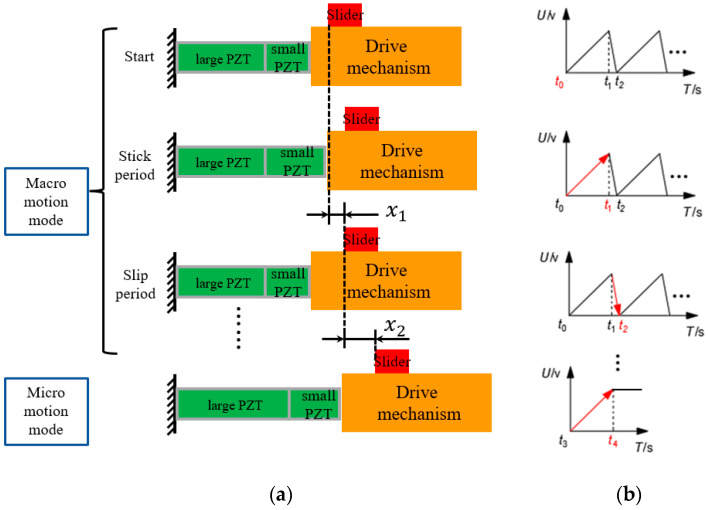
(**a**) Principle of dual piezoelectric stick-slip driving and (**b**) conventional driving signal.

**Figure 2 micromachines-13-02008-f002:**
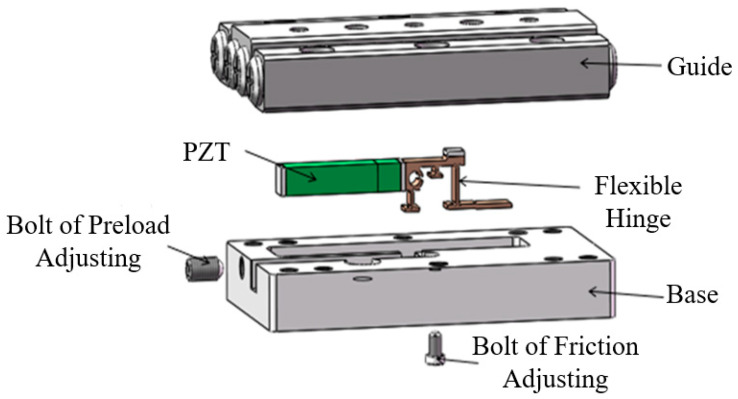
Structural model diagram of the proposed positioning stage.

**Figure 3 micromachines-13-02008-f003:**
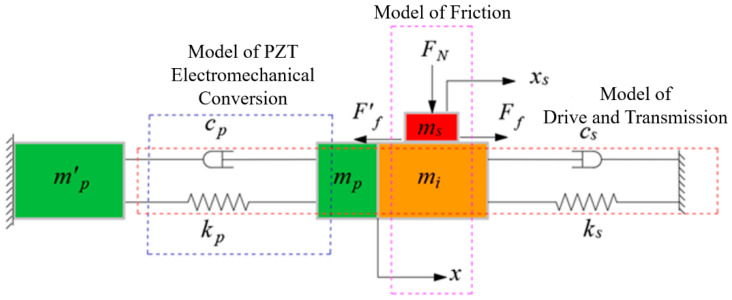
Dynamic model of the nanopositioning stage.

**Figure 4 micromachines-13-02008-f004:**
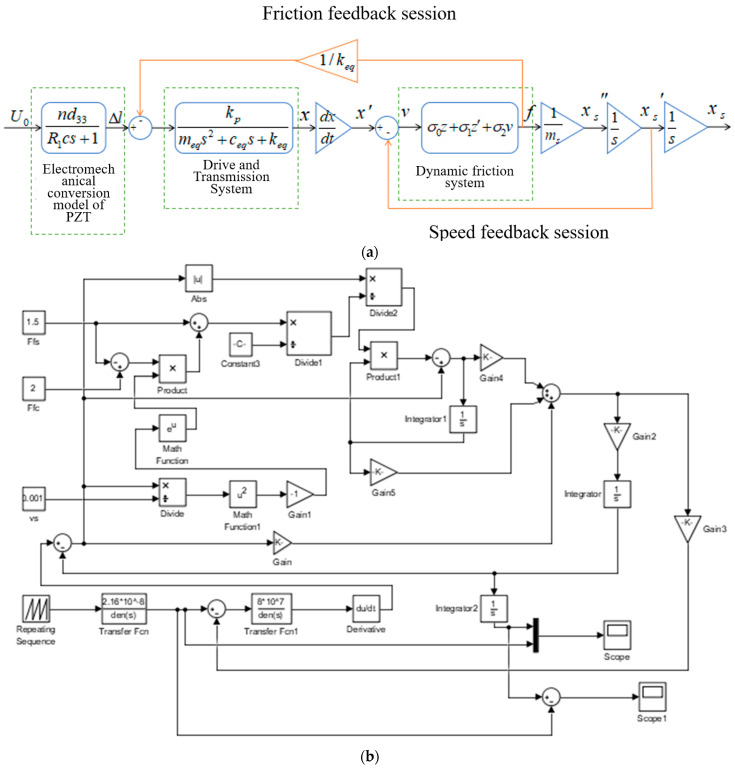
(**a**) Nanopositioning stage system dynamics model transfer flow chart and (**b**) simulation model of the nanopositioning stage.

**Figure 5 micromachines-13-02008-f005:**
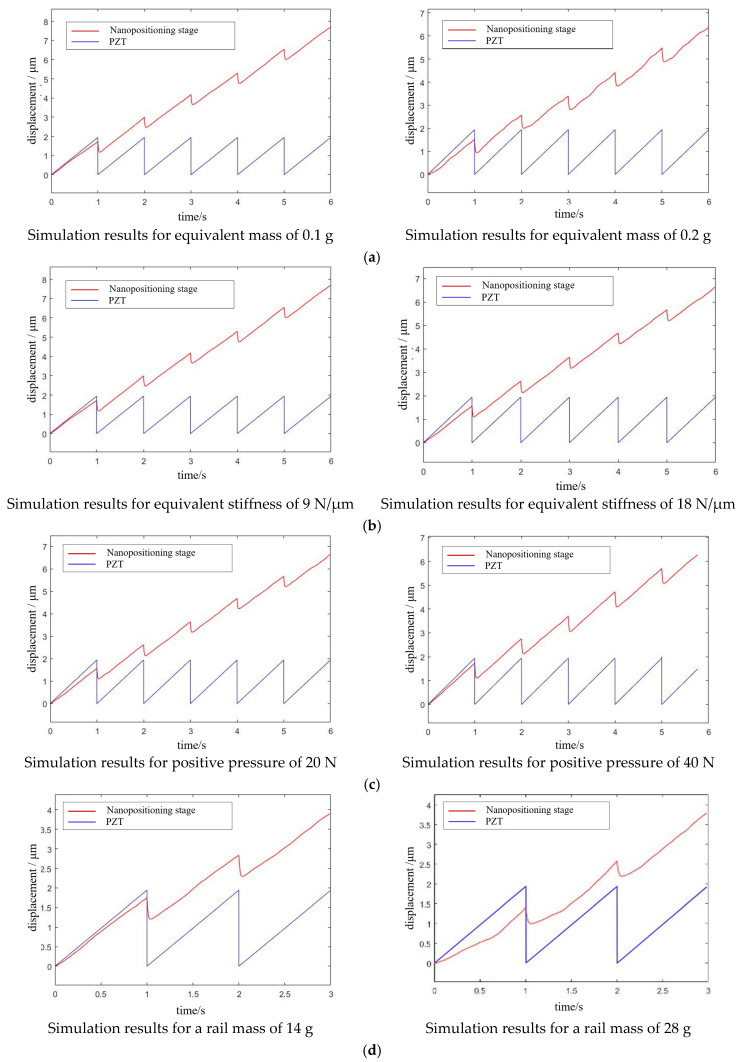
Effect of different parameters on the output displacement of the positioning stage: (**a**) effect of change in equivalent mass of flexible hinge; (**b**) effect of change in equivalent stiffness of flexible hinge; (**c**) effect of change in positive pressure providing friction; and (**d**) effects of changes in the quality of the guide rails.

**Figure 6 micromachines-13-02008-f006:**
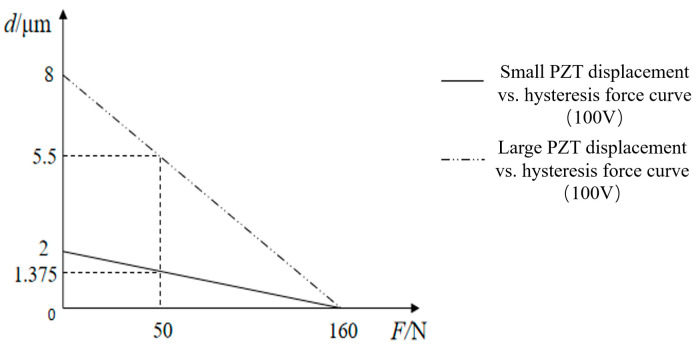
PZT displacement vs. hysteresis force curve.

**Figure 7 micromachines-13-02008-f007:**
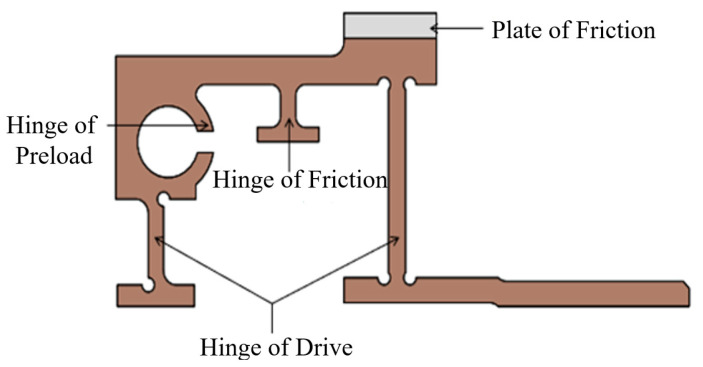
Zoom in hinges.

**Figure 8 micromachines-13-02008-f008:**
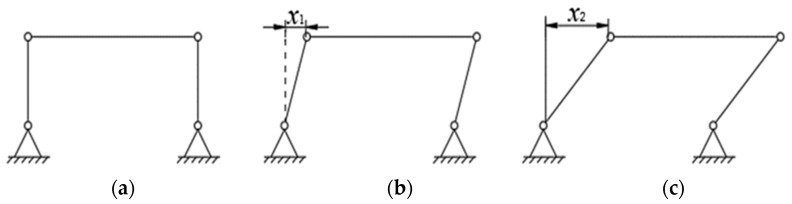
Deformation principle of flexible transmission parts. (**a**) System initial state sketch. (**b**) Sketch of system increasing preload state. (**c**) System output state sketch.

**Figure 9 micromachines-13-02008-f009:**
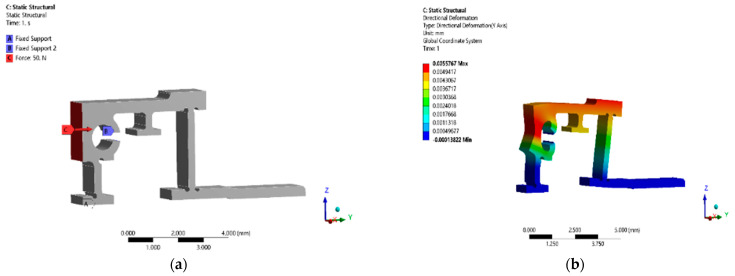
Simulation process (**a**) and results (**b**) of flexible hinge displacement.

**Figure 10 micromachines-13-02008-f010:**
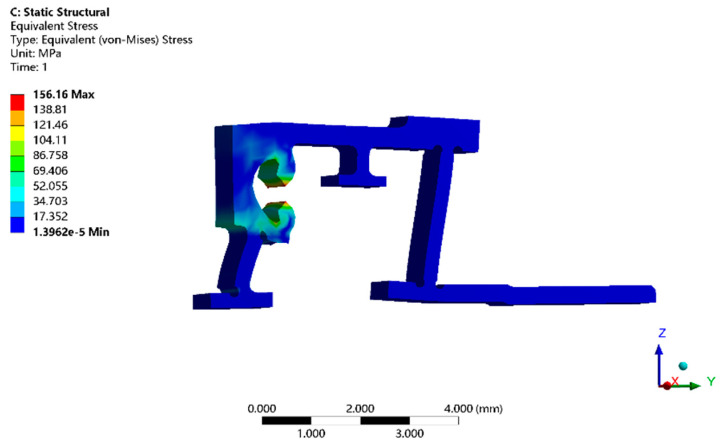
Flexible hinge stress simulation results.

**Figure 11 micromachines-13-02008-f011:**
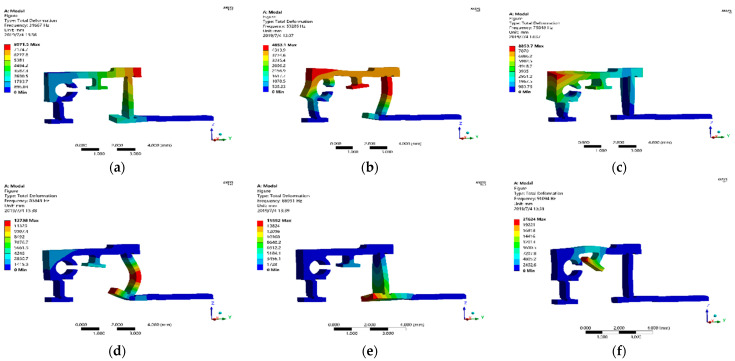
Simulation results of flexible hinge modal analysis (**a**–**e**) represented from first-order mode to sixth-order mode, respectively.

**Figure 12 micromachines-13-02008-f012:**
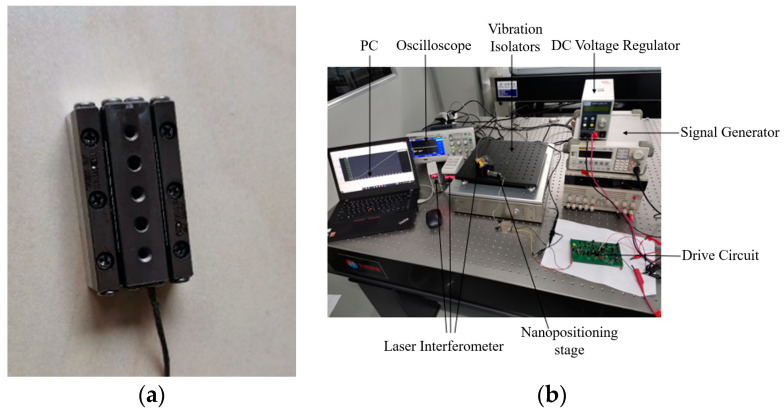
(**a**) Constructed stage and (**b**) experimental system for performance measurement.

**Figure 13 micromachines-13-02008-f013:**
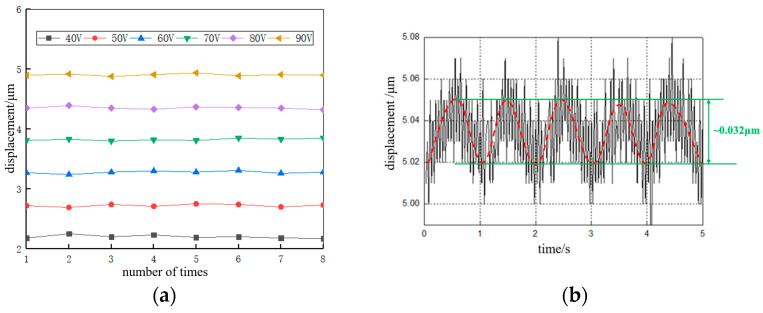
The results of the “micro-motion” function test. (**a**) Micro-motion state range-of-motion experimental results and (**b**) experimental results of micro-motion state motion resolution.

**Figure 14 micromachines-13-02008-f014:**
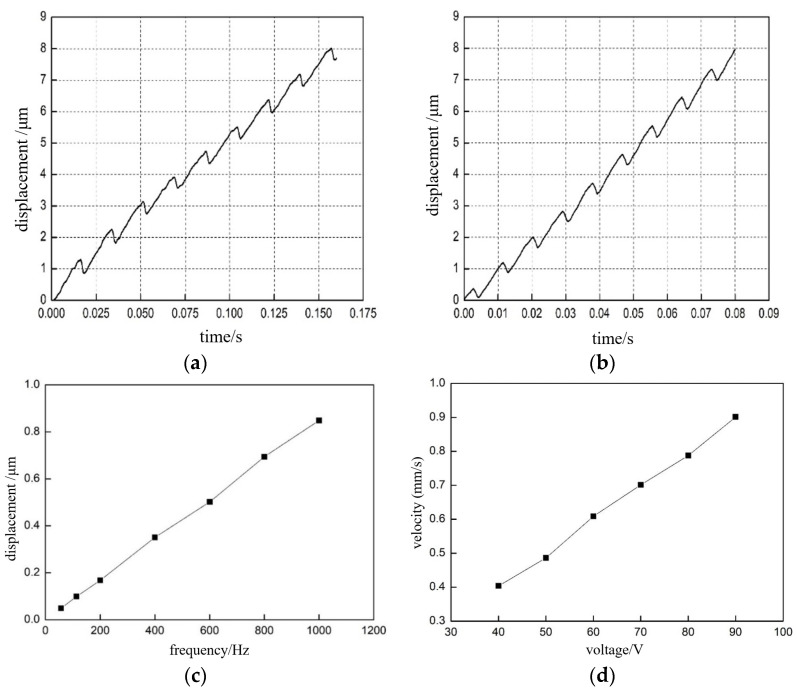
Experimental results of the effect of different input parameters on the motion state of the “macro-motion”: (**a**) positioning stage displacement output curve when the driving frequency is 50 Hz; (**b**) positioning stage displacement output curve when the driving frequency is 100 Hz; (**c**) stage average displacement output curve, driven by multiple drive frequencies; and (**d**) average stage displacement output curve driven by multiple drive voltages.

**Figure 15 micromachines-13-02008-f015:**
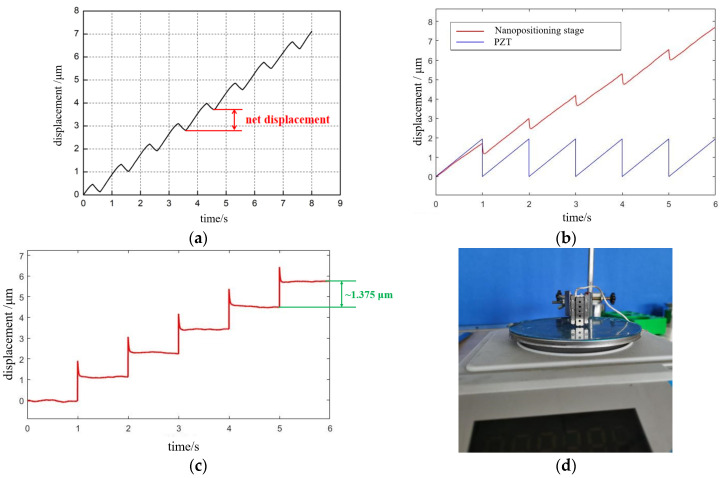
The results of the “macro-motion” function test: (**a**) experimental results of the maximum step length of the macro-motion; (**b**,**c**) validation of simulation results of simulation methods; (**d**) positioning stage maximum thrust test experiment; (**e**,**f**) experimental results of the minimum step size of the macro-motion; and (**g**,**h**) consistency experiment of forward and backward motion of positioning stage.

**Figure 16 micromachines-13-02008-f016:**
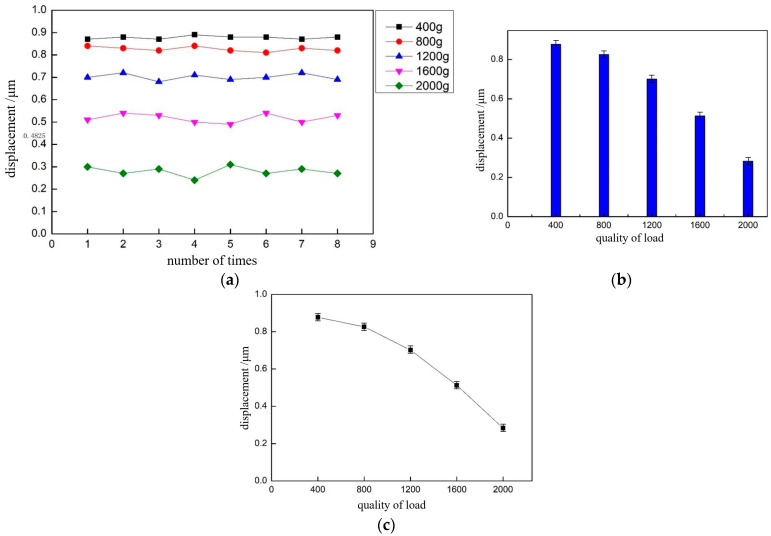
Output displacement of the positioning stage under different loads. (**a**,**b**) The relationship between output displacement of nanopositioning stage and load quality. (**c**) The relationship between the forward motion speed of nanopositioning stage and the load quality.

**Table 1 micromachines-13-02008-t001:** Characteristic parameters of PZT.

Type	Dimension	Nominal Displacement	Maximum Output Force	Stiffness	Static Capacity	Resonant Frequency (kHz)
	(mm^3^)	(μm)	(N)	(N/μm)	(nF)	
Large PZT	2 × 2 × 8	8	160	20	88	115
Small PZT	2 × 2 × 2	2	160	80	22	115

**Table 2 micromachines-13-02008-t002:** Positioning stage output displacement data at different loads.

Load Quality & Serial Number	400 g (μm)	800 g (μm)	1200 g (μm)	1600 g (μm)	2000 g (μm)
1	0.87	0.84	0.70	0.51	0.27
2	0.88	0.83	0.72	0.54	0.29
3	0.87	0.82	0.68	0.53	0.24
4	0.89	0.84	0.71	0.50	0.31
5	0.88	0.82	0.69	0.49	0.27
6	0.88	0.81	0.70	0.54	0.29
7	0.89	0.83	0.72	0.50	0.27
8	0.88	0.82	0.69	0.53	0.28
Average value	0.8775	0.8263	0.7013	0.5175	0.28
Error	0.0070	0.0106	0.0146	0.0198	0.0378

## Data Availability

Not applicable.

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
