# Peer review of "A Novel Monopolar Cross-Scale Nanopositioning Stage Based on Dual Piezoelectric Stick-Slip Driving Principle"

_micromachines, 2022, doi:10.3390/mi13112008_

Round 1

Reviewer 1 Report

The authors have addressed all questions before publication in this journal.

1) Some figures are not clear, such as Fig. 4 and 5.

2) In Figure 5, plafor should be changed to platform?

3)P9 lines 280, Mpa changed to MPa, please double check units.

4) In Figure 14, the simulated results should be added to compare with experimental results in order to verify the simulation method.

5) The resolution in this manuscript should be compared with the published results to demonstrate its high performance.

Reviewer 2 Report

This manuscript proposes a dual piezoelectric stick-slip driving principle for a cross-scale Nanopositioning Stage. This two piezoelectric elements stick-slip driving mechanism is expected to have many applications.

This manuscript effectively demonstrates the stick-slip driving design, simulation and experiment results. The information of nanopositioner performance such as  maximum motion range of 20 mm, minimum step length of 70 nm, maximum scanning range of 4.9 μm, motion resolution of 16 nm, maximum motion speed of 10mm/s and maximum load of 2 kg are also disclosed, giving the reader very useful information. 

Therefore, the reviewer considers this paper to be worthy of publication in Micromachines when some issues below are solved: 

1. In Figure 5 d, why the simulation results show that the heavier 28g rail have similar step size to 14g one? Stick slip mechanism works better when the rail or driven mass is larger.

2. There are some errors:

Figure 1. What is the vertical axis on the right nand side t0, t1, t2?

A macro-micro dual-drive high-acceleration, high-precision XY positioning stagestage is proposed by Jie D et al.

Figure 5. Nanopositioning platfor

3. As long as the magnetic field is closed, it is possible to use magnets inside SEM. There is one very similar stick-slip nanopositioning system (https://doi.org/10.1016/j.ohx.2022.e00317) relevant to this study. They can achieve atomic resolution, nanoscale step size and centimeter range. It’s better to mention in the introduction section.

Reviewer 3 Report

The article has some smaller and bigger issues which should be addressed.

- It will be very helpful if the authors explain the abbreviations used the first time they occur (e.g. voice coil motor (VCM), etc.)

- Figure 1(b) micro motion mode, from the graph, is not clear that a constant voltage is applied, and explicitly not between t3 and t4 (maybe after t4, but this is not drawn)

- Figure 3; Some variables in Figure 3 are not explained in the text. 

- Figure 4; It will be very suitable when authors write the equations according to which drawn the Simulink scheme. From the model, it is not possible to say, if it is the model of the presented nanopositioning stage.

- Figure 5; In the description of 5 (a) are stated equal values for "equivalent mass of 0.2g"

- Figure 5; On the y-axes, there are values in micrometers, but in each graph is a multiplier x10^(-6), which means that dependencies are in picometers

- Figure 5; it will be great if the dependencies will be somehow  differentiated (not only by color), consider different line styles, or add markers (in the case of greyscale print the dependencies can not be distinguished)

-Figure 5; Only speculation. Is your legend correct? I am afraid, that blue dependence is the output displacement of PZT and red is the output displacement of the nanopositioning platform. 

- Figure 9 (a) Please, consider redrawing the location of Fixed Support (mainly 2) and the place of the Load. (b) If the Load is acting on the whole surface "C" (according to (a)), how can the such surface be deformed (according to (b))? Note: If it is your Figure 9 (b) correct then, the stiffness is 50N/5.5767µm = 8.966N/µm and not 9.1N/µm as you stated. 

- line 276; They are serial-connected two PZT with nominal displacements of 2µm and 8µm, why do you consider only 8µm displacement of the piezoelectric ceramic?

- please resolve the conflict between the statement on line 341 and Figure 14 (d).

- line 379; could you explain more your statement: "The speed ... is good." what it means? Have you made your statement/conclusion only from one measurement with a frequency 50Hz? There are achieved equal results of speed for different frequencies in full range (you declare frequencies up to 15kHz)? 

- Figure 15 (e), (f); could you replace the x-label with "time/s"?

- lines 396 to 411 output displacement; could you explain the maximum motion range of the proposed positioning stage? In the abstract is a value of 20mm, unfortunately, I do not know according what you have made this conclusion. In this part of the paper, you made measurements with different loads, but the highest displacement is only 0.89µm. Could you more focus to explain this part, what means measured displacements and their influence on the whole motion range of the stage?

- lines 420 - 422; can you somehow substantiate your statement that the dimensions of the nanopositioning stage have been significantly reduced, some comparison with other nanopositioning stages? 

- line 423; could you add some reference/link to the figure or highlight in the article according to what you got the resolution of 16nm in micro-motion mode?

-line 424; could you replace "maximum motion range" with "maximum micromotion range"

- Conclusion; could you add information about the maximum motion range of the proposed positioning stage to the Conclusion too?

Round 2

Reviewer 1 Report

It can be accepted.

Reviewer 3 Report

Thanks to the authors for their replies and changes in the paper. I think, that paper is well interesant and should be published without changes.

I have only one remark.

In Fig. 4 (b),  in a few blocks of Gain and Constant is not clear what is set (MATLAB it automatic changes to -K-), maybe it would be great if you change the title of the block to the name of used a variable (but this is only advice for better understanding of the simulation model)